# Comparing Motor-Evoked Potential Characteristics of NEedle versus suRFACE Recording Electrodes during Spinal Cord Monitoring—The NERFACE Study Part I

**DOI:** 10.3390/jcm12041404

**Published:** 2023-02-10

**Authors:** Maria C. Gadella, Sebastiaan E. Dulfer, Anthony R. Absalom, Fiete Lange, Carola H. M. Scholtens-Henzen, Rob J. M. Groen, Frits H. Wapstra, Christopher Faber, Katalin Tamási, Marko M. Sahinovic, Gea Drost

**Affiliations:** 1Department of Neurosurgery, University Medical Centre Groningen, University of Groningen, 9713 GZ Groningen, The Netherlands; 2Department of Anaesthesiology, University Medical Centre Groningen, University of Groningen, 9713 GZ Groningen, The Netherlands; 3Department of Neurology, University Medical Centre Groningen, University of Groningen, 9713 GZ Groningen, The Netherlands; 4Department of Orthopaedics, University Medical Centre Groningen, University of Groningen, 9713 GZ Groningen, The Netherlands; 5Department of Epidemiology, University Medical Centre Groningen, University of Groningen, 9713 GZ Groningen, The Netherlands

**Keywords:** intraoperative neurophysiological monitoring, muscle recorded transcranial electrical stimulation motor evoked potentials, subcutaneous needle recording electrode, surface recording electrode, volume conduction, signal-to-noise ratio

## Abstract

Muscle-recorded transcranial electrical stimulation motor-evoked potentials (mTc-MEPs) are used to assess the spinal cord integrity. They are commonly recorded with subcutaneous needle or surface electrodes, but the different characteristics of mTc-MEP signals recorded with the two types of electrodes have not been formally compared yet. In this study, mTc-MEPs were simultaneously recorded from the tibialis anterior (TA) muscles using surface and subcutaneous needle electrodes in 242 consecutive patients. Elicitability, motor thresholds, amplitude, area under the curve (AUC), signal-to-noise ratio (SNR), and the variability between mTc-MEP amplitudes were compared. Whereas amplitude and AUC were significantly higher in subcutaneous needle recordings (*p* < 0.01), motor thresholds and elicitability were similar for surface and subcutaneous needle recordings. Moreover, the SNRs were >2 in more than 99.5% of the surface and subcutaneous needle recordings, and the variability between consecutive amplitudes was not significantly different between the two recording electrode types (*p* = 0.34). Surface electrodes appear to be a good alternative to needle electrodes for spinal cord monitoring. They are non-invasive, can record signals at similar threshold intensities, have adequately high SNRs, and record signals with equivalent variability. Whether surface electrodes are non-inferior to subcutaneous needle electrodes in detecting motor warnings is investigated in part II of the NERFACE study.

## 1. Introduction

Intraoperative neurophysiological monitoring (IONM) has become the gold standard for spinal cord monitoring during spinal surgery [1,2,3,4,5]. Transcranial electrical stimulation motor-evoked potentials (Tc-MEPs) are used to monitor the motor pathways [5], and somatosensory-evoked potentials (SSEPs) are used to monitor the sensory pathways [6]. Tc-MEPs can be recorded directly in the epi- or subdural space over the spinal cord using direct-waves (D-waves) or from the muscles (mTc-MEPs). 

mTc-MEPs can be recorded using either extramuscular (surface or subcutaneous needle) or intramuscular (needle or hookwire) electrodes [5,7,8]. When mTc-MEPs are recorded with extramuscular electrodes, volume conduction significantly affects the recorded potentials by influencing the waveform morphology, decreasing the measured amplitudes, and changing the frequency content of the signals [9,10]. 

Electromyography (EMG) and intraoperative motor root studies have shown that spontaneous discharges, e.g., myotonic discharges, fibrillations, and positives spikes, cannot be adequately registered using surface electrodes [11,12]. Moreover, Skinner et al. concluded that EMG monitoring during myelopathy surgery should not be performed using surface electrodes [12]. It was stated that intramuscular electrodes are preferred for lower motor neuron function monitoring, since near field recordings are required. However, which recording electrodes are preferred for upper motor neuron function monitoring, i.e., spinal cord monitoring, has not been vigorously investigated yet. 

Different mTc-MEP characteristics are considered of importance during spinal cord monitoring. These include mTc-MEP elicitability, motor threshold, amplitude, area under the curve (AUC), signal-to-noise ratio (SNR), and the variability of mTc-MEP amplitudes over time [5]. Two studies have compared mTc-MEP amplitudes of signals recorded using intramuscular versus extramuscular electrodes [12,13]. Both found that higher amplitudes were recorded with intramuscular electrodes. To the best of our knowledge, no human studies have examined the effects of two different extramuscular mTc-MEP recording methods, i.e., the use of surface and subcutaneous needle recording electrodes, on mTc-MEP characteristics. 

When supramaximal stimulation is applied, an mTc-MEP amplitude reduction of ≥50–80% is most often used as a warning criterion of impending neurological damage during spinal cord monitoring [5]. The polysynaptic, nonlinear, and unstable properties of mTc-MEP monitoring signals are associated with a higher incidence of false warnings [14]. To avoid excessive numbers of false warnings, it is important that the variability of the mTc-MEP signals is as small as possible. The difference in variability between successively recorded mTc-MEP amplitudes is unknown for different mTc-MEP recording electrodes in humans. 

Therefore, the aim of this study was to compare mTc-MEP characteristics, including elicitability, motor thresholds, amplitudes, AUC, SNR, and the variability of mTc-MEP amplitudes, between surface and subcutaneous needle electrode recordings, during mTc-MEP spinal cord monitoring of the tibialis anterior (TA) muscles.

## 2. Materials and Methods

### 2.1. Study Design

This is a prospective observational study. Since the data were collected during routine clinical care, the hospital ethical committee waived the requirement for full ethical committee review following the terms of the Dutch Act on Medical Research on Human Subjects (Wet Medisch-Wetenschappelijk Onderzoek, or ‘WMO’). The study was, however, registered and approved by a non-WMO study evaluation committee who deemed informed consent unnecessary.

### 2.2. Patients

Consecutive patients of 12 years and older who underwent surgery with mTc-MEP spinal cord monitoring were included. Subcutaneous needle and surface recording electrodes were placed on the TA muscles in all patients. Patients in whom warning criteria were reached were excluded from this study. A warning was defined as a reproducible significant deterioration or complete loss of mTc-MEP amplitude of the TA left and/or right muscles. The warning criteria depended on the type of surgery and were either ≥50% or ≥80% deterioration of mTc-MEP amplitude [14]. Measurements from muscles that were not elicitable and subjects who had fewer than 10 mTc-MEP measurements were also excluded. Muscles where the mTc-MEPs could not be elicited consistently, i.e., where the mTc-MEPs could be elicited one time and not the next, were only included for the elicitability analysis. 

### 2.3. Anesthesia 

Anesthetic management was at the discretion of the responsible anesthesiologist and in keeping with the departmental protocols. In brief, anesthesia was achieved and maintained with infusions of propofol with either remifentanil or sufentanil [15]. Propofol administration was titrated to achieve a bispectral index (Medtronic, Ireland) of 40–60 (representing adequate but not excessive depth of anesthesia). Muscle relaxants were given prior to tracheal intubation but not thereafter to avoid negative effects on mTc-MEPs. During surgery, esketamine was used as an analgesic at the discretion of the anesthesiologist in 101 patients (54.59%). The responsible anesthesiology teams attempted to maintain arterial blood pressure within 30% of baseline, and the core temperature, oxygen, and carbon dioxide partial pressures within normal range during surgery. Inhalational anesthetics were not used. 

### 2.4. Muscle-Recorded Transcranial Electrical Stimulation Motor Evoked Potentials

#### 2.4.1. Stimulation Parameters

Intraoperative mTc-MEPs were evoked using a constant voltage stimulator (NIM-Eclipse E4 IONM system, Medtronic BV, Eindhoven, The Netherlands). Transcranial electrical stimuli were administered with corkscrew electrodes montaged at stimulation location Cpl1–Cpl2 (1 cm posterior, 1 cm lateral) altered from the international 10–20 EEG-system [16]. Stimulation was performed using a train of 5 pulses and a pulse duration of 75 µs. Four patients were monitored with a pulse duration of 0.5 ms and 2 patients were monitored with >5 pulses per train. For each patient, mTc-MEPs were measured using different interstimulus intervals (ISI) of 1 ms, 1.25 ms, 1.5 ms, 2 ms, 3 ms, and 4 ms to determine the optimal setting for baseline. The ISI that provided the highest amplitudes was selected for monitoring during the surgical procedure. A high-pass filter of 30 Hz and a low-pass filter of 1500 Hz were applied. 

#### 2.4.2. Recording Method

mTc-MEPs were recorded using subcutaneous needle and surface electrodes at the left and right TA muscle, as shown in Figure 1. A bipolar montage and true differential amplification were applied to surface and needle recordings. The skin underneath the surface electrodes was scrubbed briefly prior to the electrode placement. 

The following recording electrodes were used during the study:Subcutaneous needle electrode: 13 mm length × 0.40 mm width (27 G), noncoated, straight, (Medtronic, Xomed, Jacksonville, FL, USA);Surface electrode: 20 × 27 mm, adhesive surface pad electrodes (Medtronic, Xomed, Jacksonville, FL, USA).

One surface electrode was placed at the junction of the upper one-third and lower two-thirds of the line between the tibial tuberosity and the tip of the lateral malleolus. The second surface electrode was placed over the lateral aspect of the tibia 4 cm distal to the first recording electrode (muscle belly–tendon preparation). The subcutaneous needle electrodes were inserted into the skin directly under the surface electrodes at an angle of 45 degrees, after which tape was placed over the surface and subcutaneous needle electrodes to avoid detachment.

#### 2.4.3. Elicitability

Both TA muscles in all patients were checked for elicitability. For all muscles with inconsistently elicitable mTc-MEP responses, the number of responses and non-responses for both recording methods were calculated. For the other mTc-MEP characteristics investigated in this study (motor threshold, amplitude, AUC, SNR, and variability), the TA muscles with inconsistently elicitable mTc-MEP responses were excluded from the analysis. 

#### 2.4.4. Motor Threshold

Motor thresholds were determined at the beginning and the end of the surgery. The motor threshold was determined by increasing stimulation voltage in predefined steps of 10–20 V. The motor threshold was defined as the lowest voltage that generated a reproducible mTc-MEP amplitude at a display gain of 50 µV. 

#### 2.4.5. Amplitude and Area under the Curve

After baseline measurements had been completed, the first mTc-MEP measurement that was performed after positioning of the patient but before incision was collected for the left and right TA muscle. The mTc-MEP amplitudes and AUCs recorded by the subcutaneous needle and surface electrode electrodes were compared. 

To analyze the difference between the beginning and end amplitudes, the mean of the first 3 amplitudes was divided by the mean of the last 3 amplitudes per patient, type, and side.

#### 2.4.6. Signal-to-Noise Ratio (SNR)

Signal-to-noise ratios were calculated for the first 50 included patients. Each mTc-MEP amplitude was divided by the largest noise amplitude. The noise amplitude was calculated from the last part of the signal (Figure 2). Differences in geometric mean SNR were compared for surface and subcutaneous needle mTc-MEP recordings. 

#### 2.4.7. Variability

To assess the variability between mTc-MEP amplitudes for both surface and subcutaneous needle recording electrodes, all consecutive, simultaneous amplitudes during surgery were collected. Previous results have shown that needle electrodes on average record larger mTc-MEP amplitudes than surface electrodes; the mean consecutive difference (MCD) is not considered a useful measure of variability since the absolute difference would be larger for the needle electrode [17]. Therefore, consecutive ratios of the amplitudes were calculated per patient, per side, and per electrode type, and a geometric mean of these consecutive ratios (MCR) was calculated for each subgroup (patient, side, type) using Formula (1).
(1)$$\mathrm{MCR}=\sqrt[{n-1}]{\left|\frac{m_{2}}{m_{1}}-1\right|*\left|\frac{m_{3}}{m_{2}}-1\right|*\ldots *\left|\frac{m_{n}}{m_{n-1}}-1\right|}$$

In this formula, *m*_1_, *m*_2_, etc. are the individual amplitudes and *n* is the number of measurements. 

### 2.5. Data Collection 

mTc-MEP curves were exported from the NIM-Eclipse E4 IONM system (Medtronic BV, The Netherlands) after which the mTc-MEP amplitudes, AUCs, and noise amplitudes were calculated and collected using software routines written in Python (version 3.7.1.). The consecutive mTc-MEP amplitudes were plotted per patient, per muscle and per type of recording electrode for visual inspection to objectify the elicitability. Repetitive mTc-MEP measurements (every 10 s) and artifacts were excluded from the analysis of the SNR and MCR. Motor thresholds were collected from the neurophysiologist IONM reports for each patient, electrode type, and TA muscle. 

### 2.6. Statistical Analysis

All analysis were performed in R software version 4.0.5 (the R foundation for statistical computing). The mTc-MEP parameters, including elicitability, motor threshold, amplitude, AUC, amplitude difference, SNR, and the variability between mTc-MEP amplitudes were compared between surface and subcutaneous needle recordings. Descriptive statistics and patient-wise histograms were used to identify potentially influential values and outliers. Normally distributed variables were summarized as mean and SD, while non-normally distributed variables were summarized as median and interquartile range (IQR).

The interclass correlation coefficient (ICC) was calculated to identify the agreement of the percentage of not-elicitable responses between surface and subcutaneous needle recording electrodes (icc function in R version 4.0.5). The ICC calculation was based on the responses of all the included TA muscles (*n* = 364). 

For all other outcomes, the best-fit linear regression model was identified by testing whether the type of electrode (subcutaneous needle vs. surface), side (left vs. right), and their interaction significantly improved model fit (*p* < 0.05) using sequential likelihood ratio tests (Anova function in R, version 4.0.5). Surgery time was an additional variable that was tested for model fit improvement of the mTc-MEP amplitude difference and variability. Model diagnostics were performed on the best-fit model. Sensitivity analyses were performed to verify the robustness of the results to potentially influential values and outliers by refitting the models without those values. After generating the linear regression models, the residuals were plotted to see if they were normally distributed. If they were not normally distributed, the variable was log-transformed using the natural logarithm.

## 3. Results

### 3.1. Patients

A total of 242 consecutive patients were identified of whom 185 were included in the study. Among the remaining 185 patients, twelve had inconsistent elicitable mTc-MEP responses in at least one TA muscle. Overall, responses from 182 left TA muscles and 182 right TA muscles were included in the analysis (see Figure 3). Baseline patient characteristics are presented in Table 1.

### 3.2. mTc-MEP Parameters

#### 3.2.1. Elicitability

From the 185 included patients, a total of twelve patients had inconsistently elicitable responses, six patients had inconsistently elicitable responses on the left, two had inconsistently elicitable responses on the right, and five had them on both TA muscles (Table 2). All muscles that had elicitable responses had 0% not elicitable responses for both the surface and the subcutaneous needle electrode.

The ICC was calculated to identify the agreement of the percentage of not elicitable responses between surface and subcutaneous needle recording electrodes. The agreement between electrode types was excellent for the left (ICC > 0.999, 95% CI [0.999, 1.00]) and right TA muscles (ICC = 0.999, 95% CI [0.998, 0.999]), and it was greater than would be expected by chance (*p* < 0.001).

#### 3.2.2. Motor Threshold

From the 179 patients, motor thresholds were missing in one patient for the needle electrode since the needle electrode was placed after threshold measurements. The mean voltage threshold was 183.60 V (SD = 55.93) for the left and 176.59 V (SD = 55.47) for the right surface electrode recordings, and it was 183.22 V (SD = 56.48) for the left and 176.23 V (SD = 56.33) for the right subcutaneous needle electrode recordings at the start of surgery. At the end of surgery, the mean threshold was 185.08 V (SD = 58.66) for the left and 181.48 V (SD = 60.84) for the right surface electrode recordings, and it was 184.32 V (SD = 58.82) for the left and 180.82 V (SD = 63.26) for the right subcutaneous needle recordings (Figure 4). The thresholds were 1.92 V (SD = 30.37) higher for the left and 5.14 V (SD = 30.57) for the right surface electrode recordings, and they were 1.54 V (SD = 31.28) higher for the left and 4.28 V (SD = 31.95) for the right subcutaneous needle recordings at the end compared to the start of surgery.

Multiple linear regression was used to test if electrode type and side significantly predict motor thresholds. At the start of surgery, there was no difference between motor thresholds and recording electrode type (surface or subcutaneous needle recording electrodes) (adjusted R2 < 0.01, F(1, 704) < 0.01, *p* = 0.93, 95% CI [−7.72, 8.66]) or side (right or left) (adjusted R2 < 0.01, F (1, 704) = 2.78, *p* = 0.10, 95% CI [−15.27, 1.27]). At the end of surgery, the motor thresholds were also not significantly different between the types of electrodes (adjusted R2 <0.01, F(1,703) = 0.02, *p* = 0.87, 95% CI [−8.21, 9.64]) and side of the recording (adjusted R2 <0.01, F(1, 703) = 0.61, *p* = 0.43, 95% CI [−12.48, 5.37]).

#### 3.2.3. Amplitude, Area under the Curve and Amplitude Difference

The median amplitude (of the first measurement after baseline for each patient) of the TA left and right were 413.89 µV (IQR [177.32, 894.45]) and 443.12 µV (IQR [158.12, 900.85]) for the surface electrode recordings and 703.51 µV (IQR [295.95, 1410.46]) and 707.66 µV (IQR [279.62, 1684.96]) for the subcutaneous needle electrode recordings, respectively. The median AUC of the TA left and right was 2046.25 (IQR [784.32, 4853.67]) and 1873.62 (IQR [784.32, 4853.76]) for surface recordings and 2723.26 (IQR [1153.88, 6348.76]) and 3007.94 (IQR [1009.00, 6282.58]) for subcutaneous needle recordings. The median amplitude difference of the TA left and right were 1.12 (IQR [0.90, 1.17]) and 1.06 (IQR [0.83, 1,58]) for the surface electrode recordings and 1.22 (IQR [0.87, 1.69]) and 1.10 (IQR [0.83, 1.55]) for the subcutaneous needle electrode recordings.

Amplitudes, AUCs and amplitude differences were log-transformed since the residuals were not normally distributed. Significantly lower log-transformed amplitudes were found for surface electrode recordings compared to subcutaneous needle recordings (adjusted R2 = 0.03, F(1, 696)= 24.53, *p* < 0.01). Log-transformed amplitudes and side of the recordings were not significantly associated (adjusted R2 < 0.01, F (1, 696) = 0.29, *p*= 0.59, 95% CI [−0.24, 0.14]). mTc-MEP amplitudes were 47.15% lower (95% CI [−0.66, −0.28]) when recorded with the surface electrodes compared to subcutaneous needle electrodes. The log-transformed AUC was significantly higher for the subcutaneous needle recording electrode recordings (adjusted R2 < 0.01, F(1, 696) = 6.77, *p* < 0.01). AUCs were 26% lower (95% CI [−0.45, −0.06]) when recorded with the surface electrode. No significant difference was found between AUC and side of the recording (adjusted R2 < 0.01, F (1, 696) = 0.08, *p* = 0.78, 95% CI [−0.22, 0.17]). No significant difference was found between the log amplitude differences and the type (adjusted R2 <0.01, F(1, 696) = 0.02, *p* = 0.89, 95% CI [−0.10, 0.08]) or side (adjusted R2 < 0.01, F(1, 696) = 3.16, *p* = 0.08, 95% CI [−0.17, 0.01]) of the recordings. Moreover, no significant effect of surgery duration on the log amplitude difference was found (adjusted R2 < 0.01, F(1, 694) = 0.31, *p* = 0.58, 95% CI [−0.02, 0.03]).

#### 3.2.4. Signal-to-Noise Ratio

Signal to noise ratios (SNRs) were calculated for the first 50 patients. The SNR of the left and right TA was >2 in 2026/2035 (99.6%) and 2026/2037 (99.5%) for surface recordings and 2033/2044 (99.5%) and 2041/2050 (99.6%) for subcutaneous needle recordings, respectively. The mean of the geometric mean SNRs for the left and right TA muscles was 39.58 (SD 24.16) and 37.44 (SD 27.09) for surface recordings and 53.09 (SD 39.39) and 53.89 (SD 36.39) for the subcutaneous needle recordings, respectively. The surface electrode recordings were associated with 14.98 lower geometric mean SNR compared to subcutaneous needle recordings (adjusted R2 = 0.05, F(1, 198) = 10.8, *p* < 0.01, 95% CI [−23.97, −5.99]). The SNR was not associated with the side of the electrodes (adjusted R2 > −0.01, F(1, 198) = 0.02, *p* = 0.89, 95% CI [−9.90, 8.56]).

#### 3.2.5. Variability

A total of 9886 left and 9488 right surface and 9881 left and 9503 right subcutaneous needle mTc-MEPs recordings were used to calculate MCRs in 179 patients. The mean MCR of the left and right TA was 0.20 (SD 0.11) and 0.19 (SD 0.10) for surface electrode recordings and 0.20 (SD 0.12) and 0.21 (SD 0.12) for subcutaneous needle recordings. The geometric mean consecutive ratios for both recording electrodes per side are shown in Figure 5.

The influence of recording electrode type, side, and surgery time per patient were analyzed with multiple linear regression. Electrode type (adjusted R2 < 0.01, F(1, 696) = 0.92, *p* = 0.34, 95% CI [−0.02, 0.01]) and side (adjusted R2 <0.01, F(1, 696) <0.01, *p* = 0.95, 95% CI [−0.02, 0.02]) were not significantly associated with the variability of the mTc-MEP amplitudes. Surgery time per patient (adjusted R2 < 0.01, F(1, 694) = 4.41, *p* = 0.04, 95% CI [<0.01, 0.01]) was significantly associated with the geometric mean consecutive ratio. Per hour longer surgery duration, the MCR increased by 0.0053 (Figure 6).

## 4. Discussion

In this study, we compared the characteristics of mTc-MEPs recorded from the TA muscles with surface and subcutaneous needle recording electrodes during spinal cord monitoring. Although amplitudes and AUC were significantly higher when recording mTc-MEPs with subcutaneous needle electrodes compared to surface recordings, in all other aspects, the surface electrode recordings were equivalent. There was an almost perfect agreement in elicitability rating between the surface and the subcutaneous needle electrode mTc-MEPs. Furthermore, the motor thresholds were similar for surface and subcutaneous needle recording electrodes. Although the SNR was significantly higher with subcutaneous electrodes, in all recordings, the SNR was sufficiently high, making it possible to distinguish signals from noise for both recording electrode types. Lastly, the variability between mTc-MEP amplitudes is similar for both recording electrodes.

Individual factors such as sweat content and subcutaneous fat content influence the volume conduction of the measured surface signal. The interaction of volume conduction with the basic source characteristics can only be evaluated using biophysical modeling [18]. The extra impedance of the skin and subcutaneous tissue, which must be overcome when recording with surface electrodes, might influence the volume conduction. However, the volume conduction for both recording methods does not seem to influence the ability to record an mTc-MEP, since thresholds necessary to elicit an mTc-MEP of the TA muscle were similar when recording with either surface or subcutaneous needle recording electrodes. Furthermore, the ability to detect mTc-MEPs in muscles that showed inconsistently elicitable mTc-MEP responses was similar for both recording methods.

The amplitudes were significantly higher when recording mTc-MEPs with subcutaneous needle electrodes compared to amplitudes recorded with surface electrodes. This corresponds with the intra-operative findings of other studies [13,19]. Gonzalez et al. showed that recordings from intramuscular electrodes have significantly higher amplitudes than subcutaneous electrodes [13]. Journee et al. found higher amplitudes with subcutaneous electrodes compared to surface electrodes in horses [19]. The AUC was also significantly higher with subcutaneous needle recordings compared to surface mTc-MEP recordings.

The amplitude differences were not significantly different between the subcutaneous needle and surface recordings. In addition, the amplitude differences were not significantly influenced by the total surgery time. Therefore, the time of surgery seems to affect mTc-MEPs recorded with both electrode types equally.

When mTc-MEP monitoring is performed, the reproducibility of the recordings is of great clinical importance. Therefore, the variability between mTc-MEP amplitudes should be as low as possible [5]. Not only will this decrease the number of false-positive warnings, but a consistently lower variability might also lead to stricter and more precise warning criteria [14,20]. Since there are significant differences in height of the amplitudes between the recordings of the two different electrode types, the coefficient of variation (calculated from mean and SD) and MCD (calculated from absolute amplitudes) were not suitable measures of variability [17]. Therefore, in this study, we used the MCR as a measure of variability. The mean variability was similar for surface and subcutaneous needle electrodes and below 0.25 in most patients. This is considered an acceptable variability for mTc-MEP monitoring, since in our study, ≥50% or ≥80% amplitude reduction criteria were used. The total surgery time did have an effect on variability of the mTc-MEPs. A possible explanation is the likelihood of more variation in anesthetic and surgical influences (such as blood pressure, body temperature and anesthetic infusion rates), with increasing time. The effect of 0.0053 increase in MCR per hour is small, and further research is needed to validate the clinical relevance.

One could argue that impedances are usually higher for surface electrodes than for subcutaneous needle electrodes. However, this study showed sufficient SNRs for mTcMEPs measured with both recording electrodes.

An advantage of surface recording electrodes is that they are non-invasive, and thus, there is no risk of infection, hemorrhage, damage to surrounding tissue, or needle-stick injury [21,22,23]. In the hands of experienced IONM personnel, the placement of surface electrodes is as time-consuming as subcutaneous needle electrode placement.

As discussed in the introduction, surface electrodes are considered inferior to intramuscular electrodes for identifying spontaneous discharges in lower motor neurons [12,13]. However, during spinal cord monitoring, surface electrodes are a good alternative when motor root monitoring is not required. We have investigated whether surface electrodes are non-inferior to subcutaneous needle electrodes for the detection of motor warnings in the NERFACE study part II.

### Limitations

Although this is one of the most extensive human studies comparing surface with subcutaneous needle electrodes for intraoperative mTc-MEP monitoring, only one surface electrode and subcutaneous needle electrode size was used. No human studies were found that investigated the effects of electrode size during mTc-MEP monitoring. Larger surface electrodes can obtain signals from a larger part of the muscle and could therefore have an influence on the variability of the signal. Van Dijk et al. studied the effect of electrode size on compound muscle action potential (CMAP) variability in 20 healthy subjects and found that the area of the surface recording electrode was inversely related to the variability of the CMAP amplitudes [24]. Therefore, mTc-MEP characteristics, especially the variability, obtained from other electrode sizes could be different from the results presented in this study.

Furthermore, we did not objectively determine if the subcutaneously placed needle electrodes were indeed placed subcutaneously instead of intramuscularly (e.g., in underweight patients with little subcutaneous fat). Inadvertent intramuscular placement could have had effects on volume conduction. Although intramuscular recording needles acquire signals from fewer muscle fibers than subcutaneous recording needles, Skinner et al. showed that intramuscular recordings are superior in monitoring the peripheral motor neuron [12]. This can be explained by spontaneous muscle fiber activity which can be better detected using intramuscular recording electrodes [12,13]. However, if the needle electrodes were indeed placed intramuscularly in some cases, it would only enforce our conclusion that volume conduction does not greatly influence the motor threshold measurements for mTc-MEPs. Moreover, waveform analysis was not performed, so no conclusions regarding the similarity of the waveform characteristics other than AUC can be made. Other aspects of the waveform, such as polyphasia and length of the signal, could be influential to the visual interpretation of the recorded mTc-MEPs. This could be relevant since, in theory, polyphasia could be reduced by the pathological loss of motor units [5]. Lastly, this study evaluated the quality of mTc-MEP responses with recording electrodes on the TA muscle. For application of these results in mTc-MEP monitoring of other muscles, one should consider that the TA muscle is a relatively superficial muscle. Therefore, dissimilar results could be found in the surface recordings of deeper muscles.

## 5. Conclusions

Surface recording electrodes have advantages over subcutaneous needle recording electrodes for spinal cord mTc-MEP monitoring, since they are non-invasive and obtain signals with similar elicitability, thresholds and variability. SNRs and amplitudes are higher for subcutaneous needle electrodes, but nevertheless, they are sufficiently high for spinal cord monitoring and interpretation with surface electrode mTc-MEP recordings.

## Figures and Tables

**Figure 1 jcm-12-01404-f001:**
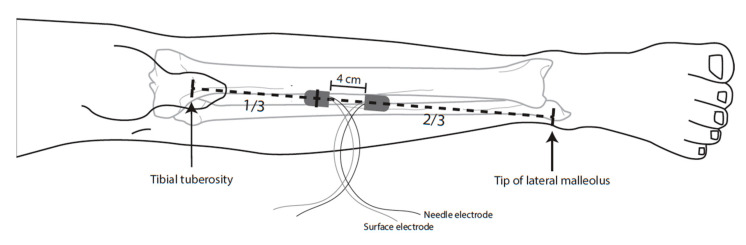
Placement of surface and subcutaneous needle recording electrodes on the tibialis anterior muscle of the right leg.

**Figure 2 jcm-12-01404-f002:**
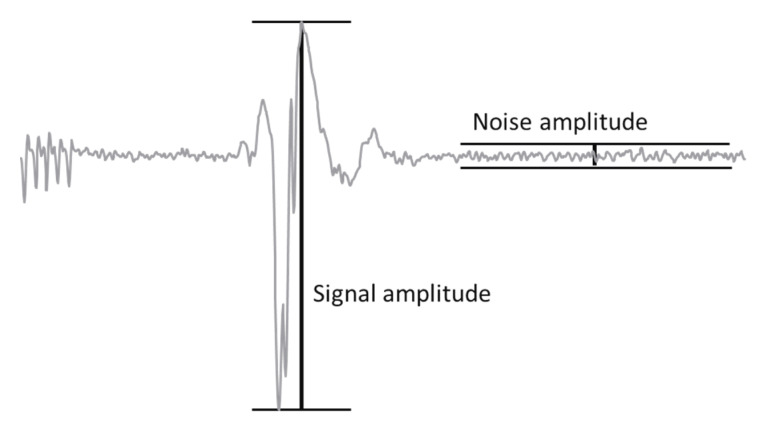
Schematic presentation of an mTc-MEP amplitude and a noise amplitude.

**Figure 3 jcm-12-01404-f003:**
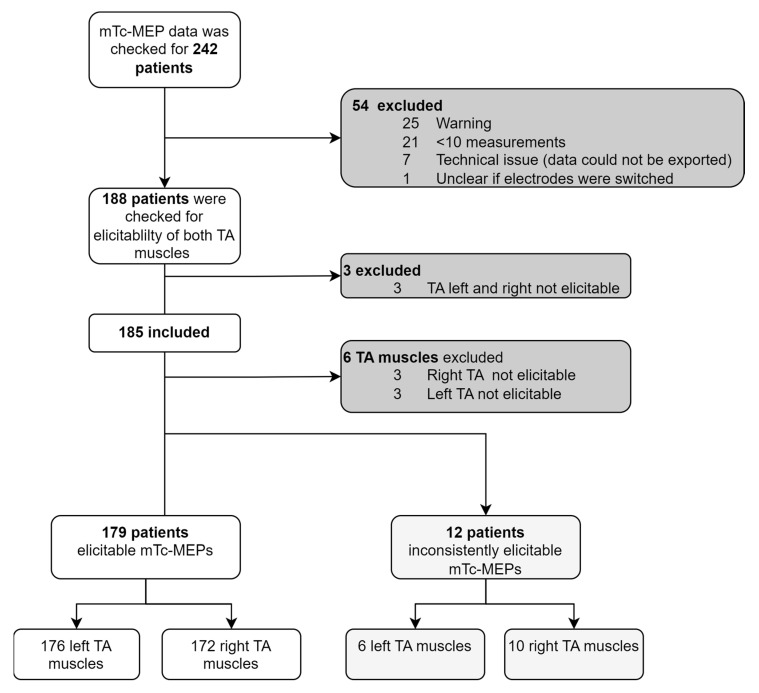
Flowchart patient selection. TA, tibialis anterior; mTc-MEP, muscle transcranial electrical stimulation motor evoked potential.

**Figure 4 jcm-12-01404-f004:**
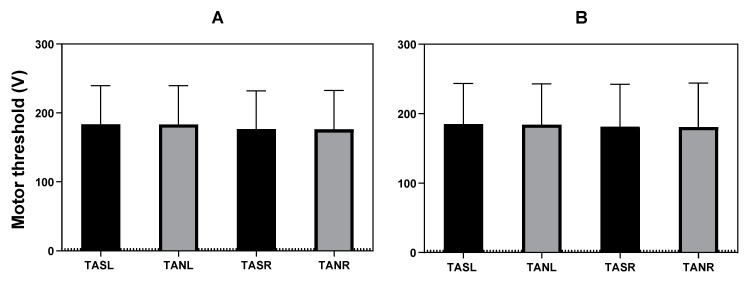
Motor thresholds at the start (**A**) and end (**B**) of surgery. The *x*-axis denotes the different muscles and recording electrodes used for the mTc-MEP monitoring, and the y-axis denotes the mean motor thresholds in volt. The mean and SD (whiskers) of the motor thresholds for the surface (black bars) and subcutaneous needle electrodes (gray bars) are shown for both the motor thresholds at the start of surgery (**A**) and end of surgery (**B**). Mean motor thresholds were not significantly different between electrode type and side both at the start and end of surgery. TASL, tibialis anterior surface left; TANL, tibialis anterior subcutaneous needle left; TASR, tibialis anterior surface right; TANR, tibialis anterior subcutaneous needle right.

**Figure 5 jcm-12-01404-f005:**
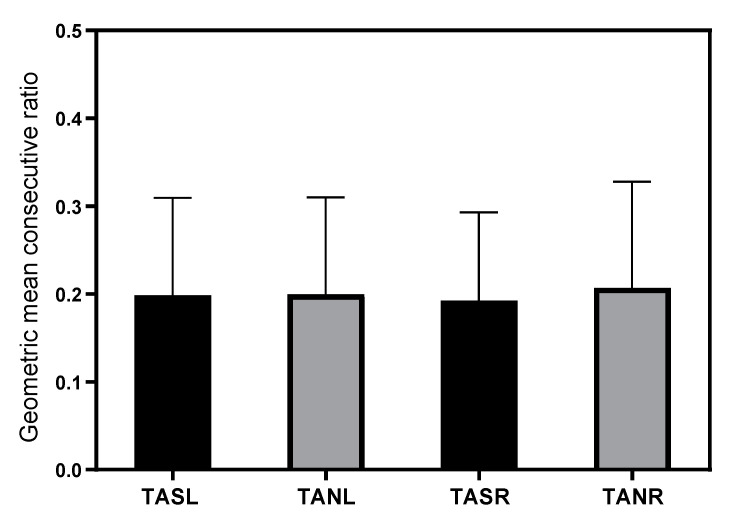
Effects of recording electrode type on geometric mean consecutive ratio of mTc-MEP amplitudes. The x-axis denotes the different muscles and recording electrodes used for mTc-MEP monitoring, and the y-axis denotes the geometric mean consecutive ratio. The mean MCR and SD (whiskers) of the mTc-MEP amplitudes for the surface electrodes (black bars) and subcutaneous needle electrodes (gray bars) are shown. Geometric mean consecutive ratios were not significantly different between electrode type and side. TASL, tibialis anterior surface left; TANL, tibialis anterior subcutaneous needle left; TASR, tibialis anterior surface right; TANR, tibialis anterior subcutaneous needle right.

**Figure 6 jcm-12-01404-f006:**
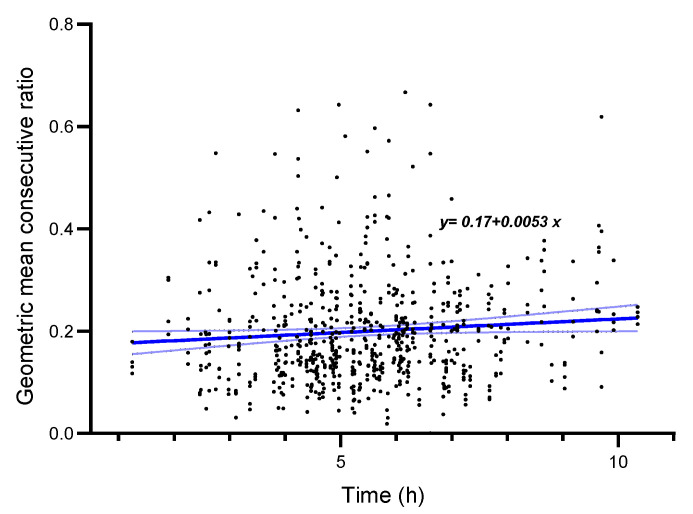
Effect of surgery time on geometric mean consecutive ratio of mTc-MEP amplitudes. The *x*-axis denotes the time of surgery in hours, and the y-axis denotes the geometric mean consecutive ratio. The blue line is the fitted linear regression line, and the corresponding formula of the line is stated in the figure. Time of surgery in hours was significantly associated with the geometric mean consecutive ratio (*p* = 0.04).

**Table 1 jcm-12-01404-t001:** Baseline patient characteristics.

Characteristics	Patients n = 185
**Median age in years (IQR)**	18 (15–46)
**Female N (%)**	120 (64.9)
**Diagnosis N (%)**	
Orthopedic surgery	115 (62.2)
Idiopathic scoliosis	82 (64.9)
Syndromic scoliosis	13 (7.0)
Neuromuscular scoliosis	16 (8.7)
Congenital scoliosis	1 (0.5)
Kyphosis	3 (1.6)
Neurosurgery	54 (29.2)
Intramedullary tumor	12 (6.5)
Intradural extramedullary tumor	21 (11.4)
Intradural cauda equina tumor	9 (4.9)
Thoracic HNP with spinal cord compression	3 (1.6)
Transdural	1 (0.5)
Extradural	2 (1.1)
Tethered spinal cord	3 (1.6)
Spinal nerve root tumor	3 (1.6)
Transdural	1 (0.5)
Extradural	2 (1.1)
ATSCH	1 (0.5)
Trauma with spinal cord compression (extradural)	1 (0.5)
Degenerative spine instability (extradural)	1 (0.5)
Vascular surgery (endovascular aneurysm surgery)	16 (8.7)
**Mean surgery time in minutes (SD)**	326.21 (100.8)
**TA strength prior to surgery N (%)**	
**Right**	
MRC < 5	20 (10.8)
MRC = 5	165 (89.2)
**Left**	
MRC < 5	21 (11.4)
MRC = 5	164 (88.67)
**TA strength after surgery N (%)**	
**Right**	
MRC < 5	17 (9.2)
MRC = 5	168 (90.8)
**Left**	
MRC < 5	18 (9.7)
MRC = 5	167 (90.3)

TA, tibialis anterior; MRC, medical research council scale; HNP, herniated nucleus pulposus; ATSCH, anterior thoracic spinal cord herniation.

**Table 2 jcm-12-01404-t002:** Percentage of not elicitable responses per patient with inconsistently elicitable responses in at least one muscle.

Patient	% Not Elicitable TASL	% Not Elicitable TANL	% Not Elicitable TASR	% Not Elicitable TANR
1			25.0	25.0
2			18.2	18.2
3			79.3	79.3
4			75.7	75.7
5			13.1	13.1
6			9.9	11.6
7	13.7	13.7		
8	29.0	32.3		
9	19.6	19.0	20.7	21.2
10	74.1	77.8	68.5	68.5
11	7.8	7.8	15.5	26.2
12	78.1	78.1	71.9	71.9

TASL, tibialis anterior surface left; TANL, tibialis anterior subcutaneous needle left, TASR; tibialis anterior surface right; TANR, tibialis anterior subcutaneous needle right.

## Data Availability

The data may be available at a reasonable request.

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
