# Peer review of "Comparing Motor-Evoked Potential Characteristics of NEedle versus suRFACE Recording Electrodes during Spinal Cord Monitoring—The NERFACE Study Part I"

_jcm, 2023, doi:10.3390/jcm12041404_

Round 1
Reviewer 1 Report
This is an interesting and valuable study, which is thoroughly planned and executed.
Major points
-
The use of Cohen’s kappa for agreement in non-elicitability between electrodes is an unusual method for intermethod comparison. Using a one-patient per row to get statistical independence, the outcome variable is not a binary or ordinal variable. The appropriate method is an intraclass correlation coefficient, comparing percent not elicitable using the two methods. A Bland-Altmann graph and calculation would also be useful to estimate the intermethod difference.
-
I am not certain that the statistical analysis of amplitudes between electrode types accounts for intra-patient dependencies. There is no mention of any kind of multi-level model in the statistics section, modelling intrapatient and interpatient variation. If such a multilevel model is used, it should be mentioned. For simplicity's sake, they could just analyze the difference in log-transformed median amplitudes between patients and electrodes, or even just compare the beginning and end of surgery like they did for motor thresholds. A simple t-test would then suffice for the main purpose of electrode type comparison, or a standard linear regression with one-patient-per-row for a more advanced analysis including covariates.
-
The operating room is an electrically hostile environment. An advantage of needle electrodes are lower impedances, with subsequent lower sensitivity to noise. This may be a moot point now with modern amplifiers, and the authors have thoroughly analyzed the SNR. Would it be too much to ask them to extract the actual impedances and give descriptives and a simple comparison between needles and surface electrodes? For beginners in IOM, such descriptives would allow them to benchmark against this expert group.
-
The major advantage of needles in my experience is setup time. Prepping for surface electrode application to get low impedances takes time, with longer surgery times and deeper patient hypothermia. Can the authors comment on this issue? I realize they cannot give data on total IOM setup time, comparing needles and surface electrodes, since they did both in the same patient. Was their amplifier so good that they could just “slap on” the electrodes, or did they have to scrub to get below <5 kOhm or other threshold?
-
I encourage the authors to investigate further MEP variability and polyphasia in future studies, which is a major limiting factor in MEP monitoring. They have the data and scripts available.
Minor issues
-
Typographical issue. The paragraph 3.2.2 should not be indented under table 2, but should belong with the main flow of the text, flowing with the next paragraph starting with “Multiple linear regression was used to test if electrode type and side significantly predict motor threshold”.
-
In the analysis of motor thresholds, the authors only give statistical significances in the text. It would be useful to know the mean differences in motor threshold, which are probably low and beneath any kind of threshold for clinical significance. Are F values really necessary?
-
For future studies, the authors might note that a preexcitability stimulus (2+5 pulses) gives more reproducible MEPs.
Author Response
Dear reviewer,
Thank you for the valuable feedback regarding our manuscript 'Comparing motor evoked potential characteristics of NEedle versus suRFACE recording electrodes during spinal cord monitoring. The NERFACE study part I.'
Attached please find a point-by-point response to the comments raised. The manuscript has been modified when necessary according to the feedback received.
Yours sincerely,
On behalf of all authors.

Reviewer 2 Report
Overall, the authors were comparing mTC-MEPS using surface vs subcutaneous needle electrode in the TA muscle. This was done to assess spinal integrity during various spinal surgeries. The rationale being if surface electrodes produce similar results (detect elicitability, signal to noise ratio, area under cruve etc) to subcutaneous electrode recordings, surface electrode could be used due to ease in placing electrodes and less invasive than needle electrode. The authors analyzed an extensive amount of motor units for comparison and concluded that overall besides size of response recorded, surface electrodes produce similar results seen in using needle electrodes. The rational and results are clear and well described as well as methodology. The results are incremental but provide valuable information for neurosurgeons using electrodes to assess spinal integrity during surgery.
Author Response
Dear reviewer,
Thank you for your feedback on our manuscript entitled: 'Comparing motor evoked potential characteristics of NEedle versus suRFACE recording electrodes during spinal cord monitoring. The NERFACE study part I.'
Please see the attachment for a point-by-point response.
Yours sincerely,
On behalf of all authors
